# Optimizing Irradiation Geometry in LED-Based Photoacoustic Imaging with 3D Printed Flexible and Modular Light Delivery System

**DOI:** 10.3390/s20133789

**Published:** 2020-07-06

**Authors:** Maju Kuriakose, Christopher D. Nguyen, Mithun Kuniyil Ajith Singh, Srivalleesha Mallidi

**Affiliations:** 1Department of Biomedical Engineering, Tufts University, Medford, MA 02155, USA; maju.kuriakose@tufts.edu (M.K.); Christopher.Nguyen@tufts.edu (C.D.N.); 2Research and Business Development Division, CYBERDYNE INC, 3013 AK Rotterdam, The Netherlands; mithun_ajith@cyberdyne.jp

**Keywords:** LED, photoacoustic imaging, ultrasound, 3-D printed photoacoustic probe holder, light delivery optimization, LED divergence

## Abstract

Photoacoustic (PA) imaging–a technique combining the ability of optical imaging to probe functional properties of the tissue and deep structural imaging ability of ultrasound–has gained significant popularity in the past two decades for its utility in several biomedical applications. More recently, light-emitting diodes (LED) are being explored as an alternative to bulky and expensive laser systems used in PA imaging for their portability and low-cost. Due to the large beam divergence of LEDs compared to traditional laser beams, it is imperative to quantify the angular dependence of LED-based illumination and optimize its performance for imaging superficial or deep-seated lesions. A custom-built modular 3-D printed hinge system and tissue-mimicking phantoms with various absorption and scattering properties were used in this study to quantify the angular dependence of LED-based illumination. We also experimentally calculated the source divergence of the pulsed-LED arrays to be 58° ± 8°. Our results from point sources (pencil lead phantom) in non-scattering medium obey the cotangential relationship between the angle of irradiation and maximum PA intensity obtained at various imaging depths, as expected. Strong dependence on the angle of illumination at superficial depths (−5°/mm at 10 mm) was observed that becomes weaker at intermediate depths (−2.5°/mm at 20 mm) and negligible at deeper locations (−1.1°/mm at 30 mm). The results from the tissue-mimicking phantom in scattering media indicate that angles between 30–75° could be used for imaging lesions at various depths (12 mm–28 mm) where lower LED illumination angles (closer to being parallel to the imaging plane) are preferable for deep tissue imaging and superficial lesion imaging is possible with higher LED illumination angles (closer to being perpendicular to the imaging plane). Our results can serve as a priori knowledge for the future LED-based PA system designs employed for both preclinical and clinical applications.

## 1. Introduction

Photoacoustic (PA) imaging has gained significant popularity for imaging functional and molecular information in both preclinical and clinical settings [1,2,3,4,5]. The technique involves sending light pulses (a few nanosecond pulse-width) into imaging planes that get absorbed by endogenous (e.g., hemoglobin) or exogenous (e.g., Indocyanine Green) tissue chromophores and generate acoustic waves, which can be detected by conventional ultrasound (US) transducers [2,6,7]. Based on the endogenous contrast provided by hemoglobin, PA imaging has shown promise in vascular functional imaging of human neonatal brains [8,9], malignant lesions [10,11,12,13,14], and monitoring therapies such as photodynamic therapy [1,15,16], etc. As PA imaging uniquely possesses the best properties of optical imaging (high spatial resolution, functional properties, and imaging speed) and US imaging (structural properties and penetration depth reaching tens of cm), its relevance and popularity are continuously increasing in clinical settings [2,4,17,18,19].

In PA phenomena, the acoustic pressure (**P_0_**) generated is proportional to the optical absorption coefficient (*µ_a_*, m^−1^) of the light absorber and locally available light fluence or radiant exposure (*f_0_*, Jm^−2^). This can be represented by [2,6,20,21]:(1)$$P_{0}=Г{µ}_{a}f_{0}$$
where, **Г** is the dimensionless, material thermal property dependent Gruneisen coefficient.

Light attenuates as it travels down through a material or tissue due to scattering and absorption. Moreover, for a limited aperture illumination, angle of illumination also plays an important role in defining local fluence (Figure 1). As a result, *f_0_* and thus PA signal intensity, **P_0_**, changes as a function of depth (distance from transducer or excitation source) and as a function of the illumination angle. Therefore, the optimization of light delivery is crucial for efficient PA imaging and obtaining high signal-to-noise (SNR) ratio at deeper penetration depths [22,23,24,25,26]. Specifically, for reflection mode PA imaging (transducer and light source on the same side of the sample), several studies demonstrated the dependence of irradiation angle and fiber (source)-to-transducer positioning on PA signal at various depths experimentally or through simulations [23,27,28,29,30,31]. Either unilateral or bilateral positioning of fiber bundles aligned with the nanosecond pulsed laser have been employed in these studies. For example, Haisch et al. utilized a mechanical setup that allowed unilateral illumination (one-side of the transducer) with 20–80 degrees range of motion [32]. The fiber bundle aperture size used in that study was shorter than the ultrasound transducer, which may hinder full aperture illumination of near field absorbers and can presumably suitable for intermediate to deep tissue imaging and is more suitable to image smaller lesions on the skin. In another study by Sivasubramanian et al., two fiber bundles were placed on either side of the transducer at a fixed angle. Change in the illumination angle would require changing the holder setup [33]. More recently, Sangha et al. designed a motorized system to change the bilateral light illumination in the 0°–60° range and concluded that the illumination geometry optimization is important to achieve high SNR at different depths [34]. All these studies point out that change in illumination geometry effects PA SNR at different depths and strongly indicate the need for a flexible handheld system that can deliver light at different angles depending on the depth of the lesions.

The light delivery optimization studies mentioned above were performed with spatially low diverging coherent laser sources. Though conventional lasers (e.g., Q-switched optical parameter oscillator (OPO)) can deliver the required pulse energy at various NIR wavelengths, their bulkiness, minimal-portability, and difficulty in operation prevent them from effortless usage in a clinical setting. Interestingly, nanosecond pulsed light-emitting diodes (LED) show promise in being an alternative to lasers, while offering cost-effectiveness and ease of operation has been recently proven to be successful in several studies [35,36,37,38,39]. Despite the low power of LEDs (about 3 orders of magnitude lower than the conventional Q-switched laser sources), their high pulse repetition rate (PRR) (maximum reported up to 16 kHz opposed) gives the opportunity to average several frames in real-time to achieve an SNR on par with conventional laser-based PA imaging (PAI). In addition, the large spatial divergence of LED arrays (~60°), could aid in irradiating larger sample area and potentially also provide a quasi-uniform illumination over several millimeters without a light diffuser. Given these attributes, LED-based PA systems demonstrate strong potential for clinical translation. As the use of LED’s in PA imaging is still in its infancy, it is important to characterize and optimize the light delivery strategies for better SNR at various depths.

LED array-based PAI studies so far have used a fixed orientation either in the reflection mode [40,41,42] or transmission mode [38,43]. In this study, we designed a flexible modular light delivery system for reflection mode PAI that is capable of orienting light from the LED arrays at various angles in the range of 0°–90° (Figure 1). Utilizing the flexible light delivery system, we evaluated PA image contrast in various tissue-mimicking phantoms (point targets in non-scattering and scattering liquid media, absorbing lesion under non-scattering liquid and scattering tissue such as the chicken breast) for the PA signal dependency as a function of irradiation angle and depth. We believe that our findings have an important impact on optimizing the design of LED-based PA probes and accelerate its clinical translation towards imaging both deeper and shallower lesions.

## 2. Materials and Methods

### 2.1. Photoacoustic System and Modular Arrangement for Varying Illumination Direction

#### 2.1.1. AcousticX

An LED-based photoacoustic system (AcousticX from Cyberdyne Inc., Tsukuba, Japan) with linear US transducer (7 MHz central frequency, 128 elements, 0.315 mm pitch, and 38.4 mm aperture size, elevation focus of 15 mm) and two 850 nm LED arrays (30 to 150 ns pulse width, 4 kHz maximum repetition rate, 200 µJ pulse energy for each array, 5 mm × 40 mm aperture size, 60° divergence) on both sides of the US detector was used for the experiments [44]. PA and US raw data were sampled at 40 MHz and 20 MHz, respectively, and data was reconstructed in real-time using an inbuilt Fourier-domain reconstruction algorithm of the system. For offline analysis, both PA and US data were reconstructed using a previously reported frequency domain beamforming algorithm [45]. Radiant exposure per pulse at the LED array surface is about 100 µJ/cm^2^ (200 µJ/pulse in an array area of 2 cm^2^). Given the LED source divergence of 60° and ~10.5 mm distance between the US transducer and the phantom surface to accommodate LED arrays for different angles, maximum radiant exposure at the phantom surface was estimated to be about 29 µJ cm^−2^ (for an area of about 6.93 cm^2^) from a single array at 0° LED illumination angle.

#### 2.1.2. Flexible LED Holder: Modular Design for Adjusting Irradiation Direction

Two identical modular hinge systems were designed, 3D printed and used to pivot LED arrays at different illumination directions with respect to the imaging plane. The modular LED holder consisted of three parts that were designed on Autodesk and printed using polylactic acid (PLA) on a MakerBot system. All pieces were joined together with 8–32 socket head screws. (Figure 2). Each of the modules consisted of four hinges that were attached to one another. These hinges can be adjusted or pivoted to create the required angle of illumination. Modules were attached to the US transducer on its one end and the LED arrays were gripped through the heat sink of the arrays on the pivoting end of the module, as shown in the figure. The inter LED array distance (between their adjacent edges) was about 1 cm to accommodate the US transducer. Experiments were done for 0°, 15°, 30°, 45°, 60°, 75°, and 90° angles using this modular arrangement. The LED array angles were adjusted with respect to the central axis of the US transducer using a custom-made protractor as shown in Figure 1d. During the experiments, both the US transducer and sample position were unaltered and only the LED sources were adjusted, to avoid PA intensity variations due to sample motion with respect to the US transducer.

### 2.2. Phantoms

#### 2.2.1. Graphite Pencil Lead Phantoms

A matrix of pencil leads (Graphite 2B 0.5 mm manufactured by June Gold, Bountiful, UT USA), arranged in 4 rows × 5 columns with a spacing of about 5 mm (columns) and 6 mm (rows), was constructed using two 3D printed plastic holders as shown in Figure 3a. The phantom construction with pencil lead is immersed in a container with water or 1% Intralipid (Sigma Aldrich Inc., Atlanta, GA, USA) solution (Figure 3b). The scattering coefficient of 1% intralipid solution is 1.8 mm^−1^ [46,47] close to the values reported for tumor tissue (1–2 mm^−1^) [48,49,50]. All experiments were conducted at room temperature (22 °C).

#### 2.2.2. Tissue Mimicking Phantom Containing Lesion with High Optical Absorbance

Tissue mimicking phantoms were prepared using agar powder (Sigma-Aldrich Inc., Atlanta, GA, USA). Titanium (IV) oxide, anatase powder (99.8%, Sigma-Aldrich Inc., Atlanta, GA, USA) was added to provide acoustic contrast and enhance the optical scattering properties of the agar. The preparation was done by slowly adding 1% wt./vol. of agar powder and 1% wt./vol. of TiO_2_ powder into continually stirred deionized water at ambient conditions to avoid clumps. The final solution was then heated above 80° C, above the melting temperature of agar, and exposed to a vacuum level of about 0.1 atm for 5 min to degas the solution and cooled it down to room temperature to obtain the final phantom. A cylindrical light-absorbing lesion with acoustic scatterers was prepared in a similar aforementioned method. Additionally, 0.5% wt./vol. graphite powder (<20 μm, synthetic graphite, Sigma Aldrich Inc., Atlanta, GA, USA) and 0.5% wt./vol. TiO_2_ powder were added. The concentration of the absorbing and scattering particles was chosen to mimic tumor tissue with an absorption coefficient (~0.2 cm^−1^) and reduced scattering coefficient (~10 cm^−1^ ) as previously reported in the literature [46,47,48,49,50,51].

### 2.3. Signal Analysis

#### 2.3.1. PA Intensity & Contrast to Noise Ratio (CNR) Calculation

The PA signal intensity of each pencil lead in the phantom was calculated by taking the maximum pixel value from the region of interest (ROI) around the target (white rectangle in Figure 3c). All the five laterally positioned PA intensities were then averaged to find mean and standard deviation (σ) of PA intensities corresponding to each depth location and illumination angles. Background (Bg) was calculated from an ROI below each point target (yellow rectangle in Figure 3c). It is important to note that the noise/background values were calculated from regions close to the signal ROIs. We chose the ROIs within close proximity of the signal ROI and not from regions at the corner or with only electronic noise, including reconstruction related artifacts that may be present when changing the illumination angle. To plot PA intensity changes as a function of angle in the tissue-mimicking phantom, PA signal intensities from the lesion was obtained by choosing the median PA intensities above the Bg level. Here also, similar to the pencil phantom case, Bg and σ were chosen from a region close to the lesion ROI. PA intensity was plotted in decibel (dB) with the formula:(2)$$PA\;intensity\;in\;dB=10log_{10}\left(PA\;intensity\right)$$
and the CNR was calculated using the formula:(3)$$CNR\;in\;dB=10log_{10}\left(\frac{PA\;intensity-Bg}{\sigma }\right)$$

#### 2.3.2. Divergence of the LED Source

Divergence of the LED source is an angular measure of the increase in irradiation area (and corresponding diameter or radius) with distance from the source. Laser light sources are known to have very low divergence while LED sources have high divergence. Sources with high divergence have lower radiant exposure per unit area on the target at a given depth than sources with lower divergence. These changes in radiant exposure can influence PA signal and hence it is critical to evaluate the divergence of the source and choose appropriate illumination angle to obtain maximum CNR. Assuming the LED is a line source aligned in the *X*-axis (parallel to the US transducer) while the emitted wavefronts take quasi-cylindrical shape in the imaging volume, an approximate divergence of the source in the *YZ* plane (Figure 4), in degrees can be computed using:(4)$$Divergence\;angle=2\times \left\{\tan^{-1}\left(\frac{target\;depth}{LED\;to\;Detector\;distance}\right)\right\}$$

The multiplying factor 2 is used to include both sides of the illumination plane. Target depth and LED to detector distance refer to the absorber (pencil lead) position in depth below the detector (US transducer) and its lateral distance to the source (LED array), respectively. In our experiment, the target and LED to detector distance were fixed while the source was pivoted, as shown in Figure 4b. We assume that the LED source exhibits a Gaussian spatial intensity profile and thus a Gaussian function is used to fit PA intensity vs. LED- illumination angle data to find the divergence of the LED arrays, given by
(5)$$f\left(x\right)=a\;e^{-{\left(\frac{x-b}{c}\right)}^{2}}$$
where *x* is the source angle, *a* is the peak PA intensity and that corresponds to angle *b*, and *c* is the half of the angle span at which PA intensity shows half maximum (50%) or −6 dB roll-off. The divergence can then be calculated by multiplying c by two, resulting in the full width of the angle at half maximum of the PA intensity (FWHM). The model can be fitted using a least-squares minimization method to obtain best-fit parameters. Moreover, the position of the peak intensity for a chosen depth can be described by
(6)d=m+n·cot(θ)
where *θ* is the LED illumination angle (Figure 1), *m* is the offset of imaging plane in depth, *n* is the separation between LED and detector, and *d* is the imaging depth.

## 3. Results

Experiments were conducted in two different phantom environments: liquid and chicken tissue. The first phantom consisted of pencil lead (point source) as absorbing targets (Figure 3) in water or 1% intralipid media. The second tissue-mimicking phantom consisted of an absorbing cylindrical lesion made of graphite powder (Figure 5), which was placed on top of a chicken breast tissue arranged obliquely to the imaging plane, while the top layer was interchanged between water or chicken breast tissue. The experiments were designed to probe the phantom at all depths simultaneously for each angle of choice, thereby reducing experimental uncertainties.

### 3.1. Pencil Lead Phantom Experiments

#### 3.1.1. Pencil Lead in Scattering and Non-Scattering Media Shows Weak Dependency on the LED Illumination Angle

Initially, PA intensities were monitored as a function of the illumination angle using a 4 × 5 pencil-lead matrix (as detailed previously) aligned orthogonal to the imaging axis to form point targets at defined locations. LED directions were varied from 0° to 90° in steps of 15°, as shown in Figure 1. Figure 3a,b shows the photographs of the lead matrix and an experimental arrangement in intralipid, respectively. PA intensity images (in dB scale) obtained at three representative angles: 15°, 45°, and 75° in water and 1% intralipid medium are shown in Figure 3c–e, Figure 3f–h, respectively. PA images from all the angles between 0° and 90° can be found in Appendix A (for water) and Appendix A (for intralipid). Angle dependent PA intensities and CNR were calculated for four depths (approximately at 12 mm, 18 mm, 24 mm, and 30 mm) as described in Section 2.3 and plotted in Figure 3i, Figure 3k, Figure 3j, Figure 3l, respectively. An approximate gap of 10.5 mm from the US transducer to the sample was kept avoiding near field reconstruction errors and also to allow room for LED adjustments. For non-scattering media (water), the PA signal intensity increased as LED illumination angle increased from parallel orientation (0°), and then reached a maximum value at intermediate angles and finally decreased in its strength as the angle approached 90° (LED illumination perpendicular to the transducer). It is obvious from the plots that the angles corresponding to the maximum PA intensity value were in an inverse relation to the target depth, by showing a maximum value at a steeper angle for the 12 mm target and a maximum value at a shallower angle corresponding for the 30 mm target.

Figure 3j demonstrates CNR obtained in water as a function of the angle of irradiation. It reveals that the PA contrast did not change significantly after reaching a maximum level for an extended angle range of 15° to 75°. This was due to an increase in the background (due to various artifacts) along with the target signal increase that in turn reduced the angle dependency. PA intensities using intralipid (Figure 3k,l) showed a similar trend as water, especially for angles between 15° to 75°. In the cases of 0° and 90°, even though the trend was similar to that of the water phantom, the changes in PA intensity were less for intralipid phantom as expected. This reduction in intensity is due to reduced fluence due to optical scattering that reduces the incoming light directionality (and thus the angle dependency) as opposed to the case of non-scattering water medium. Comparing the depth-dependent CNR within the quasi angle-independent regime (15°–75°), the total drop was larger in scattering media (~30 dB) than in the water phantom (~15 dB). This can be due to a larger attenuation promoted by increased light scattering.

#### 3.1.2. LED Source Divergence and Optimum Illumination Angle from Pencil Lead Targets in Water

Knowing the target location in depth and its lateral separation from the illumination plane, the source divergence (Figure 4) orthogonal to the illumination plane (*Y*) can be computed. In our experiments, the center of the light source (approximated here as a line source) was located at 10.5±2 mm away from the imaging plane (imaging axis of the US transducer). Utilizing Equation (5), the divergence of the LED source was calculated to be 58 ± 8° at FWHM from the Gaussian fit of PA intensities from pencil lead target at 18 mm depth (Figure 4c). The experimentally derived divergence value is in good agreement with the manufacturer’s data (60°, from Cyberdyne Inc., Tsukuba, Japan). The peak PA intensity (coefficient *b* in Equation (5)) at 18 mm depth was observed at 44° ± 3°. We further analyzed the data in Figure 3i to infer the angle at which maximum PA intensity could be obtained as a function of depth (Figure 4d) using the coefficient *b* in Equation (5). The black solid curve in Figure 4d shows the cotangent fit (Equation (6)) to the data shown in blue squares with a goodness of fit (R-squared) value equal to 0.95. Fitting was done by choosing *m* (depth offset) and *n* (separation of LED to the detector) as free fit parameters. Best fits (and 95% confidence interval bounds) obtained for, *m* is 9 mm (−0.6, 18.6 mm) and *n* is 10.7 mm (3, 18 mm). A large offset value might be due to the experimental error coming from the spatial width of the LED array, which takes up ~10.5 mm below the US transducer, in the imaging plane, at its steepest angle (90°). The value of *n* matched well to our experimentally set approximate value of 10.5 mm. From the fit, maximum PA signal intensity value at 12 mm can be obtained with a LED illumination angle of 74° while 27° can be used to image lesions at 30 mm depth. It should be noted that there exists a strong dependency of the illumination angle at superficial regions (slope of −5°/mm at 10 mm), which weakens as it goes to deeper locations (−2.5°/mm at 20 mm and −1.1°/mm at 30 mm) due to the nature of the cotangent function. The diameter of the projected beam onto the imaging plane, *Z*, at a given *θ* is estimated to be about 11.2 mm (*θ* = 84.5°) and 11.8 mm (*θ* = 27°) at 10 mm and 30 mm depths, respectively. Interestingly, less than 4% variation in the beam diameter in the imaging range is observed with different LED illumination angle. So, the large source divergence of about 11–12 mm overcomes the strong angle dependencies and can be used to lower the number of illumination angles for imaging lesions at various depths as is the case with Laser illumination. The LED illumination angle effects are even less notable in scattering medium, where deep tissue imaging can be achieved with smaller LED illumination angles (closer to being parallel to the imaging plane) and superficial lesion imaging is possible with larger LED illumination angles (closer to being perpendicular to the imaging plane).

### 3.2. Effect of Surrounding Media and Illumination Direction on PA Signal from Tumor Mimicking Lesion

A second set of experiments were conducted using tumor mimicking light-absorbing lesion placed obliquely in the imaging plane on top of chicken breast tissue (backing layer) while using water (Figure 5a) or scattering chicken tissue as the top layer (Figure 5b). This phantom study aimed to simulate superficial or deep-seated tumor lesions filled with blood vessels or contrast agents. US images corresponding to experiments using a top water layer (Figure 5c) and top chicken layer (Figure 5d) show the lesion placement and surrounding layers for comparison with corresponding PA images. The PA images using water as the top layer or chicken tissue as the top layer at LED source angles 15°, 45°, and 75° are shown in Figure 5e–g, Figure 5h–j, respectively. An apparent lateral shift in lesion positions between the images generated with water and chicken top layers was due to a change in imaging transducer placement about 2 mm in the horizontal axis, between the experiments (Figure 5h–j), which had negligible or no effects in our depth-dependent analysis.

Experiments with the top water layer show PA signals from the lesion for the entire imaging depth (Figure 5e–g), which can be associated with the negligible light scattering in water, opposed to chicken tissue (Figure 5h–j). Angle dependent PA intensity variations were also visible in both cases, where deeper tissue illumination was achieved at lower incident angles while increased PA intensity in the detector vicinity was observed for higher illumination angles. These results are very similar to those observed in Figure 3 using pencil lead phantom in water and intralipid. For a detailed analysis of depth-dependent PA intensity variation, Figure 6 was presented with analyses from three depth locations at 12 mm, 20 mm, and 28 mm, in which PA signals were plotted as a function of illumination angle, for the tissue-mimicking phantom with the top water and chicken layers. The white parallelograms in Figure 6a indicate the selected ROI from where the median PA intensity was calculated and the yellow parallelogram ROIs were considered for the background. Calculated PA intensities and CNRs as a function of angle for different depths are shown, respectively, in Figure 6b,c while using water as the top layer, and Figure 6d,f for chicken breast as the top layer. In the case of water as the top layer, the illumination angle parallel to the imaging plane, i.e., 0°, produced highest PA intensity (~23 dB) at the bottom (28 mm) than at the top regions; at 12 mm, the intensity dropped to ~17 dB as demonstrated in Figure 6b. On the other hand, experiment with the chicken top layer (Figure 6d) showed almost equal but low PA intensity (~17 dB) for all depths at 0°. It is interesting to note that the PA signal from 12 mm was quasi-constant for all angles from 30° and above. For the intermediate depth (20 mm), the intensities showed an increase up to 15°, then stayed almost unchanging until 60° and showed a slight decrease in the mean value with further increase in angle from 75° and above. It should also be noted that the signal from 28 mm depth while using chicken top layer was very close to the background level (CNR is less than 5 dB), which shows the maximum penetration depth achieved within our experimental limits.

## 4. Discussion

The dependence of image contrast on irradiation parameters such as the angle of irradiation relative to the transducer, wavelength of the light irradiation, and distance between the transducer and the light source is undisputed. In this paper, we studied the dependence of signal intensity and contrast in PA images generated by an LED light source at various illumination angles. The 3D modular hinge design gave extreme flexibility to adjust the illumination angle as well as the transducer to LED-array distance. In the current work, the distance between the LED light source and transducer was relatively constant with the LED array positioned very close to the transducer (Figure 1). Studies addressing different separation distances between the transducer and the LED array can potentially give complementary information to our results. With respect to the 3D modular hinge system itself (Figure 2), several design criteria could be improved. Currently, the footprint of the modular system is large (11.5 cm at its widest dimension). Though the complete probe is lightweight due to lack of any heavy motors or metallic pieces, it is still larger than the other 3D printed fixed angle holders, e.g., by Sivasubramanian et al. [33]. The hinge system can be further modified with computer-controlled micro-hinges, can be adapted to any fiber bundles irrespective of their shape or size and can also be integrated with any laser or pulsed diode-based systems.

In all our results, it is evident that the background signal is also increasing along with the PA signal of interest (Appendix A), resulting in less impact on CNR when the illumination angle is increased beyond a certain limit. We believe that the background signal may be affected by various artifacts like reflection artifacts, out-of-plane clutter, and side lobes. For example, it is well known that more reflection artifacts are caused by high PA signal from tissue/phantom surface reflecting off acoustically dense structures when the illumination angle is steep and the fluency is high just beneath the US probe. These reverberation type of reflection artifacts are visible in our results too (Figure 3 and Appendix A) and would have impacted the CNR calculation. At lower illumination angle setups, light can scatter outside the imaging plane, get absorbed by different features, and generate out-of-plane artifacts and this also may have an impact in the CNR [52]. We strongly believe that it is also important to consider these common artifacts in the CNR analysis hence we chose a region very close to the target of interest for the calculations, instead of choosing a blank image (image generated with no light) or electronic noise. Furthermore, we used 850 nm irradiation, a wavelength at which there is relatively high penetration depth in tissue. The utility of other illumination wavelengths will impact the PA signal intensity and CNR based on the absorption properties of lesions at those wavelengths. Our future studies will involve evaluating these observations especially in an in vivo situation with both subcutaneous (superficial lesions) and orthotopic tumors (deep lesions).

A recent study by Agarwal et al. demonstrated that single-shot laser-based PAI and LED-based PAI achieved the same SNR from lesions 2–3 cm deep in chicken tissue. Though high frame averaging (2560 frames) was performed in LED-based PAI, they achieved real-time imaging due to the high pulse repetition frequency of LED sources [53]. The utility of LED-based PAI systems has been extensively reviewed by Zhu et al. [37] and one of the key factors currently hindering the clinical translation of LED-based photoacoustic imaging systems regardless of demonstrated promise in multiple preclinical and clinical imaging applications is the optical output power of the LED arrays [37,54]. It is of paramount importance to improve the optical output power of LEDs to enhance its usage in a wide range of deep-tissue imaging applications, and thus accelerate the clinical translation. The pulse repetition rate of LEDs is several-fold higher than pulsed LASER systems and it is feasible to average multiple image frames to improve SNR without compromising on real-time imaging capability. However, averaging N frames can improve SNR by only √N, and thus this approach has its limitations. Recently several developments in beamforming methodologies are made to improve the CNR and SNR of the LED-based PA systems [55,56]. We believe that the improvement of optical pulse energy and the development of novel image reconstruction and enhancement algorithms will be critical to accelerate the clinical translation of LED-based PAI [54].

## 5. Conclusions

Our results show that the optical excitation using an LED source behaves differently than the laser excitation due to a large source divergence of LED arrays, which we calculated to be 58° ± 8°. This in turn reduces the source direction dependency of PA signal at different depths. Our analysis in the non-scattering medium shows a strong dependence of illumination angle vs. depth at near field regions (−5°/mm at ~10 mm) and weak dependence at deeper locations (−2.5°/mm at 20 mm; −1.1°/mm at 30 mm). On the other hand, results from tissue-mimicking phantom in scattering media showed significantly weaker angle dependence of PA signal intensity than the phantom in water. So, utilizing an LED-based system (or a source with similar divergence) for either deep tissue lesions or superficial lesions would be less cumbersome in terms of source alignment by offering the freedom to choose a wide range of irradiation angles without losing CNR in the images. In contrast, spatially coherent sources would require stringent alignment strategies for illuminating lesions at various depths [57,58]. The modular light delivery system and results presented in this study can serve as a priori knowledge for future LED-based PA system designs and aid in further catapulting its utility in both preclinical and clinical applications.

## Figures and Tables

**Figure 1 sensors-20-03789-f001:**
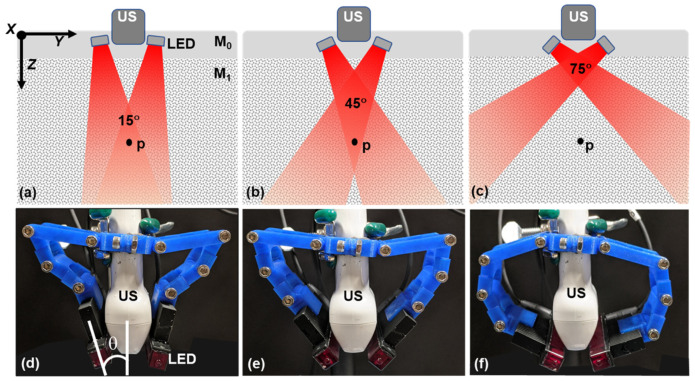
(**a**–**c**) Schematics of light emitting diode (LED) illumination cross-sectional view of photoacoustic (PA) setup at representative angles: 15°, 45°, and 75°, respectively, orthogonal to the imaging plane *XZ*, that contains a hypothetical absorber, p. M_0_ is the medium that facilitates acoustic coupling between the transducer and the phantom material M_1_; (**d**–**f**) Photographs of LED source pivoted at representative angles, *θ* = 15°, 45°, and 75°, respectively, using 3D printed modular hinge system.

**Figure 2 sensors-20-03789-f002:**
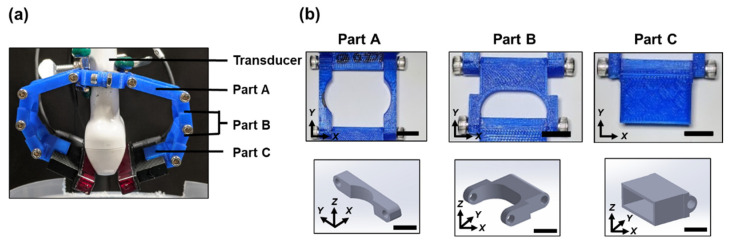
(**a**) Modular hinge system holding the LED arrays and attached to the US transducer. (**b**) The photographs of the individual pieces are shown in the top panel. The 3D renderings of the hinge pieces are shown on the bottom panel. Part A fits around the transducer and extends the horizontal reach of the holder to allow the LEDs to be placed at angles approaching 90°. Part B in conjunction with Part A allows precise horizontal and vertical height adjustment. The holder consists of two-part B pieces, and schematic of only one piece is shown in the panel. Part C holds the LEDs using the heat sinks and provides flexibility for any final adjustments on the LED illumination angle. Scale bar = 10 mm.

**Figure 3 sensors-20-03789-f003:**
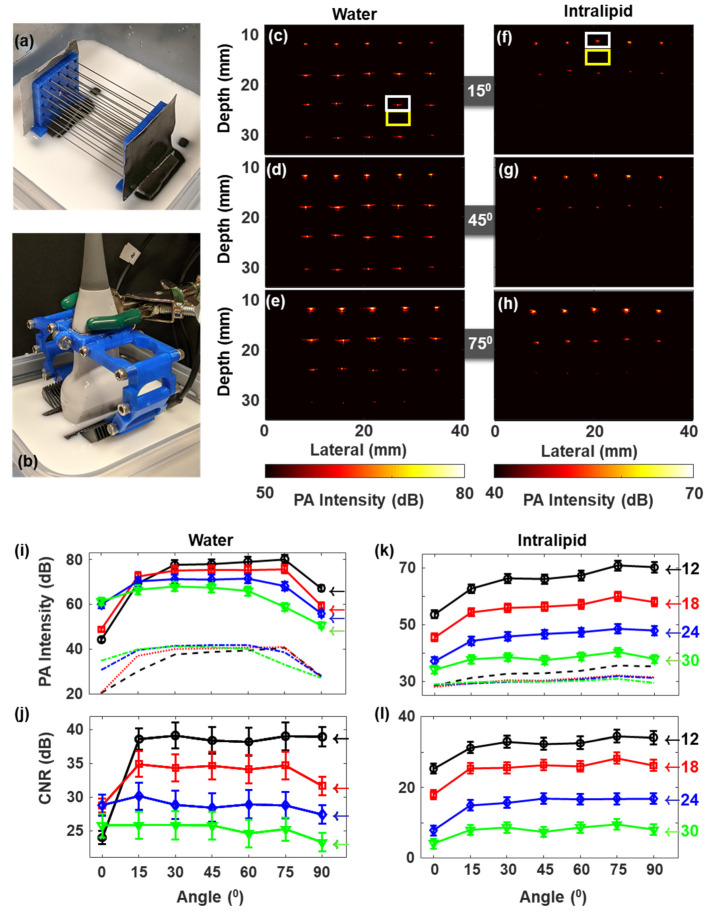
(**a**) Photograph of Pencil lead matrix; (**b**) photograph of an experimental arrangement using intralipid medium; (**c**–**h**) PA image acquired at representative angles 15° (**c**,**f**), 45° (**d**,**g**), and 75° (**e**,**h**) in water (**c**–**e**) and 1% intralipid (**f**–**h**); (**i**,**k**) Mean PA signal intensities and their standard deviations (of 5 lateral positions at each depth) plotted as a function of LED angles in water (**j**) and intralipid (**k**); and (**j**,**l**) corresponding contrast to noise ratios (CNRs) obtained as a function of LED angles in water (**j**) and intralipid (**l**). Different depths from the transducer are indicated by 12 mm (line with black circles), 18 mm (line with red squares), 24 mm (line with blue diamonds), and 30 mm (line with green downward triangles). The noise background levels corresponding to 12, 18, 24, and 30 mm are represented by black, red, blue, and green dash-dotted lines, respectively, in (**i**) in water and (**j**) in intralipid. The backgrounds were obtained right below from each signal regions, e.g., as indicated by the yellow rectangles in (**c**,**f**). Images for all the angles can be found in Appendix A (for water) and Appendix A (for intralipid).

**Figure 4 sensors-20-03789-f004:**
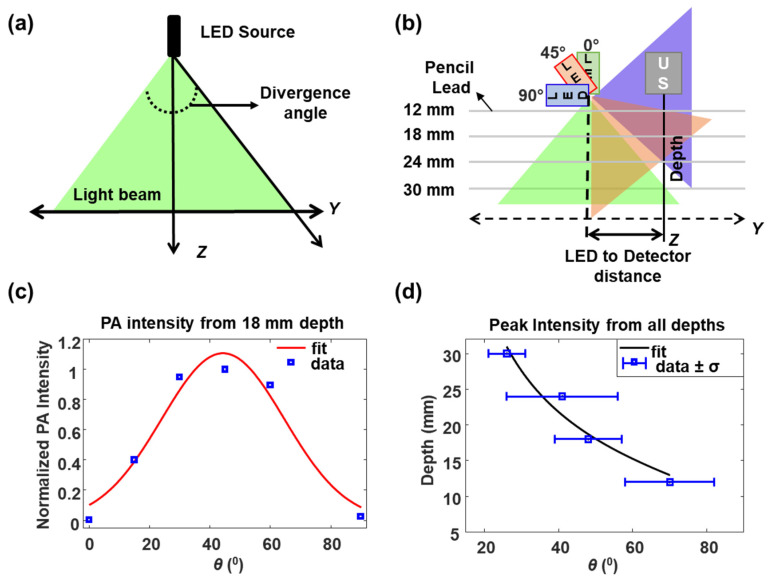
(**a**) Illustration of the source wavefront profile, while assuming the LED array as a line source with Gaussian profile having divergence described by Equation (4); (**b**) Source profile after pivoting LED arrays at three different angles in the imaging plane–green at 0°, red at 45°, and blue at 90°; (**c**) Normalized PA intensity obtained from 18 mm target vs. LED illumination angle, *θ*, is indicated in blue squares. The red solid curve shows the Gaussian fit using Equation (5), with an R-squared value of 0.95; (**d**) Blue squares with error bar show the peak PA intensity with standard deviation for each depth plotted as a function of *θ* for pencil lead phantom data in water. The black solid curve displays the best fit using a cotangent function (R^2^ = 0.95).

**Figure 5 sensors-20-03789-f005:**
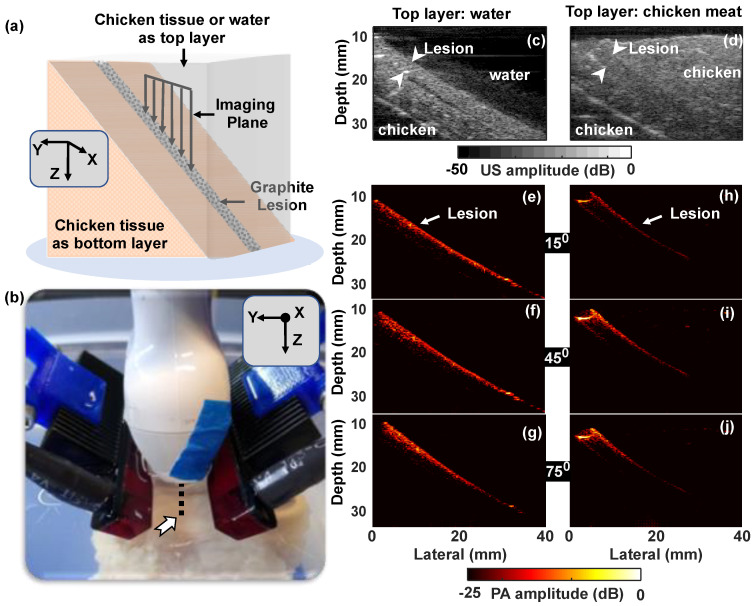
(**a**) Schematic of the tissue-mimicking phantom: Absorbing lesion (cylindrical in shape, 2 mm in diameter) placed on chicken tissue and arranged obliquely in the *XZ* plane. (**b**) Photograph of the experiment. The dotted line indicated by arrowhead shows the projection of lesion to the *x*-axis; (**c**,**d**) US image of the sample showing water top layer (**c**) or chicken layer (**d**) lesion and bottom chicken tissue in both cases; (**e**–**j**) PA intensity images captured using water as the top layer (**e**–**g**) or chicken tissue as the top layer (**h**–**j**), by choosing LED array directions: 15° (**e**,**h**), 45° (**f**,**i**), and 75° (**g**,**j**) with respect to the imaging plane.

**Figure 6 sensors-20-03789-f006:**
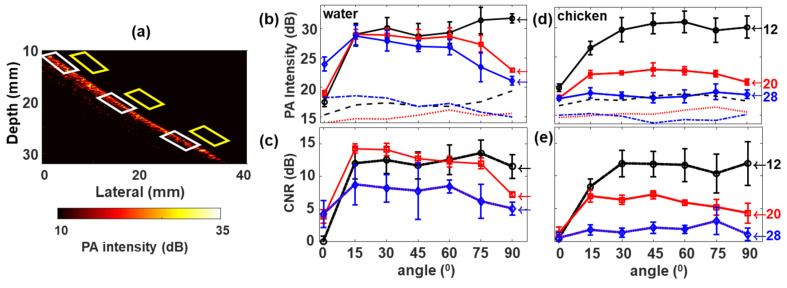
(**a**) PA image of a sample containing absorbing lesion (2 mm diameter) with the top water layer. White parallelograms indicate the regions selected for PA intensity and yellow for the background; (**b**,**d**) PA signal intensity plotted as a function of LED angles for selected depths in mm as labeled by numbers with the corresponding color, right next to each plot. Black circle with line corresponds to 12 mm, red squares with line corresponds to 20 mm and blue diamond with line indicates 28 mm. The black dotted lines, red dash-dotted lines, and blue dashed lines represent the background levels for 12, 20, and 28 mm depths (from the transducer) respectively; a gap of 10.5 mm exists between the transducer and the phantom surface. (**c**) CNR of the lesion under water; (**e**) CNR of the lesion under chicken breast. The error bars represent the standard deviation.

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
