# Peer review of "Optimizing Irradiation Geometry in LED-Based Photoacoustic Imaging with 3D Printed Flexible and Modular Light Delivery System"

_sensors, 2020, doi:10.3390/s20133789_

Round 1

Reviewer 1 Report

  • Include a section on how the LED illuminated PA angle dependency compares to laser-based PA.
  • If laser illumination can provide three orders of magnitude more energy what is the typical penetration depth (for LED illumination in chicken tissue it seems to be about 2 cm?).
  • Please discuss: Do you anticipate a large divergence angle to have a big effect for PA imaging in scattering media? Can divergence vs lack of be simulated?
  • Please discuss: Is power availability the main limitation in terms of future clinical translatability of LED photoacoustic sources?

Minor issues: 

  • General comment: need a revise for syntax, languagem=, etc:

- line 12: what is meant by sensitivity of optical imaging?

- line 17: work on sentence semantics

- line 33: space needed before citation bracket

- line 34: with “nanosecond pulse-width”

- line 36: utility? Or use of?

- line 66: space needed

- line 90: syntax

- line 108: four instead of 4

- line 114: syntax

- line 122: no need to say water is non-scattering

- line 125: gel network?

- line 136: ROI is white rectangle not square or parallelogram for fig. 2

- line 138: spelling (standard)

- line 150: homogenize?

- line 159: need a better explanation for the chicken-graphite-chicken phantom. Maybe another picture

- line 171: space between sec. 2.3

- line 192: better explanation of how divergence is computed (maybe with a picture) and why this is important to measure this in a phantom.

- line 201: describe what artifacts are seen since the PA images look clear.

- how is the angle of LED arms computed? Is there a device for calculating this?

-line 284: there is no comparison to a laser source angle dependency in this paper so maybe it should be included.

- line 287 and 291: is the preferred angle range 30-60° or 15-60°?

Author Response

  1. Include a section on how the LED illuminated PA angle dependency compares to laser-based PA.

Thank you we included it in lines: 384-399

  1. If laser illumination can provide three orders of magnitude more energy what is the typical penetration depth (for LED illumination in chicken tissue it seems to be about 2 cm?).

Based on a recent study by Agarwal et. al.,[3] LED PAI offer similar SNR at high frame rate averaged images at an experimented depth of 2-3 cm. This has been included also in the revised version. Line: 384-399

  1. Please discuss: Do you anticipate a large divergence angle to have a big effect for PA imaging in scattering media? Can divergence vs lack of be simulated?

This is a very interesting point. Consider a hypothetical situation of pencil beams emitted  from a LED and a Laser. Highly diverging beams (LED), primarily provide larger spot size on the object being imaged. Light beams with smaller divergence (Laser) have tighter spot size on the object. PA signal is primarily dependent on the optical absorption properties of the tissue and the radiant exposure at that location (light distribution is determined by both absorption and scattering properties of the tissue). Scattering in the tissue broadens the incident beam, decreasing the effective radiant exposure in the intended target area and therefore reducing the overall PA signal. Recent publications with simulations demonstrate that increasing spot size (beam width < 10 mm) enhances penetration depth[4]. With similar power output, LED array with higher divergence, therefore a broader spot size, will have lower PA signal than a source with lower divergence but greater penetration depth. If the laser beam and the LED have the same spot size, they probably will have similar penetration depth and SNR as has been demonstrated recently by Agarwal et al. [3]

  1. Please discuss: Is power availability the main limitation in terms of future clinical translatability of LED photoacoustic sources?

We included a paragraph in the discussion section. Lines 388-399

Minor issues: 

  • General comment: need a revise for syntax, language, etc:

- line 12: what is meant by sensitivity of optical imaging?

corrected

- line 17: work on sentence semantics

corrected

- line 33: space needed before citation bracket

     corrected

- line 34: with “nanosecond pulse-width”

     corrected

- line 36: utility? Or use of?

     corrected

- line 66: space needed

     corrected

- line 90: syntax

     corrected

- line 108: four instead of 4

     corrected

- line 114: syntax

     corrected

- line 122: no need to say water is non-scattering

     corrected

- line 125: gel network?

“Gel network” here refers to the phantom material. To avoid confusion, we rephrased the sentence as below in the new manuscript.

“Tissue mimicking phantoms were prepared using agar powder (Sigma-Aldrich, Inc.). Titanium (IV) oxide, anatase powder (99.8%, Sigma-Aldrich, Inc.) was added to provide acoustic contrast and  enhance optical scattering properties of the agar.”

- line 136: ROI is white rectangle not square or parallelogram for fig. 2

We corrected the discrepancy both in the text and Fig. 2 legend

- line 138: spelling (standard)

We corrected the typo.

- line 150: homogenize?

We corrected the grammatical error.

- line 159: need a better explanation for the chicken-graphite-chicken phantom. Maybe another picture

We modified the figure (Fig. 5a) to clearly explain the phantom layers and orientation. The phantom picture and also the US images in Figs 5c and 5d are labeled to show the separate layers of the phantom.

- line 171: space between sec. 2.3

Corrected

- line 192: better explanation of how divergence is computed (maybe with a picture) and why this is important to measure this in a phantom.

Thank you for the suggestion. We now included a new figure on how divergence is computed and showed further analysis on the data in section 3.1.2

- line 201: describe what artifacts are seen since the PA images look clear. 

The images look clear because the display range chosen was above the noise or background level. The same images plotted with a larger dynamic range that showcases the noise can be found in supplementary data  (Fig. S1).

- how is the angle of LED arms computed? Is there a device for calculating this?

A protractor was used to adjust angle of the LED arms. This information is added on Line 143

-line 284: there is no comparison to a laser source angle dependency in this paper so maybe it should be included.

     We edited the introduction and discussion section of the manuscript to indicate the rationale for our study and indicate the differences from the other laser source angle dependency studies (Lines 237-259 and 384-399)

- line 287 and 291: is the preferred angle range 30-60° or 15-60°?

As per the Fig.6 data on tissue mimicking phantoms,  no significant difference in CNR was observed between data obtained in the 30-75° range, hence we prefer this angle range.

References

  1. Sivasubramanian, K., et al., Optimizing light delivery through fiber bundle in photoacoustic imaging with clinical ultrasound system: Monte Carlo simulation and experimental validation. J Biomed Opt, 2017. 22(4): p. 41008.
  2. Haisch, C., et al., Combined optoacoustic/ultrasound system for tomographic absorption measurements: possibilities and limitations. Anal Bioanal Chem, 2010. 397(4): p. 1503-10.
  3. Sumit, A., et al. Photoacoustic imaging capabilities of light emitting diodes (LED) and laser sources: a comparison study. in Proc.SPIE. 2020.
  4. Ash, C., et al., Effect of wavelength and beam width on penetration in light-tissue interaction using computational methods. Lasers in medical science, 2017. 32(8): p. 1909-1918.

Reviewer 2 Report

The authors studied the angular dependency of the LED-based illumination in photoacoustic imaging. The experiments are well planed and the manuscript is written well. Although such a study was not conducted before for an LED-based system, it is well studied for laser-based illumination in a similar setting. Additionally, the findings are obvious for a light source such as LED with large beam divergence. The authors should frame the novelty of the paper and the usefulness of the finding in a better way to find acceptable for publishing.

Major: 

1) The novelty of the article is unclear. Why is this study important for the community. If the beam divergence of LEDs is known to be high what is the need to optimize the angle anyway? The only development in this work is a 2D printed adjustable LED array holder. 

2) The authors must provide optical absorption and reduced scattering coefficient of the phantom. The percentage weight does not give any indication to the reader about the scattering and absorption of the phantom. 

3) In what way is the phantom mimicking soft tissue. Authors should match both absorption and reduced scattering to well known soft tissue optical properties. How is the lesion mimicking a tumor, what optical properties are mimicked in this lesion? Authors must provide these important information otherwise the claims in this article are questionable. 

4) The discussion can be better. Readers will look for a specific angle that is optimal for the focus depth of the transducer. Add discussion on how will the findings vary with tissue having different absorption? Discuss the clinical relevance of this study. Can an adjustable light source be used in any application? Is this study helpful in any way to improve the imaging depth of LED-based PA imaging reported until now? 

Minor:

1) How is the angle measured in this 3D printed LED array holder? Is this an accurate measurement as there is no reference available in the holder.

2)  PA intensity and CNR are well-known terms and there is no need to introduce them. Authors can simply use the terms and provide the analysis.

3) Move the concept of measuring divergence to the method section and present only the results in the result section.

Author Response

We thank the reviewers for their thoughtful and valuable comments on our manuscript that have significantly improved the manuscript. We heavily revised the manuscript to incorporate the suggestions including adding more figures and supplementary data. A point-by-point response to the reviewers’ comments (in black font) are provided below. The sentences and paragraphs modified as per reviewers’ suggestions are highlighted with a line on the left margin in the manuscript document. We also refer to the new figure numbers in the response below.

Reviewer #1

1) The novelty of the article is unclear. Why is this study important for the community? If the beam divergence of LEDs is known to be high what is the need to optimize the angle anyway? The only development in this work is a 2D printed adjustable LED array holder.

LED based photoacoustic imaging (PAI)  is an emerging field with various preclinical and clinical applications. Currently a fixed angle probe (450 illumination angle) is provided with the AcousticX LED based PAI system. Though it is known that LEDs have high divergence, studies (such as ours) to understand the breadth of angle dependence in PA imaging has not been done previously. Our aim was to design a flexible holder for LEDs and understand the effect of LED illumination angle in imaging deeper (>20 mm) or shallower (close to the transducer probe) lesions. Outcome of our work shows that it is highly useful to implement larger angles  for deeper lesion imaging and smaller angles to image shallow lesions. This information is critical when designing probes based on empirical understanding of the validity of LED based systems for specific applications depending on the lesion depth. On the other hand, Lasers have low divergence and thus it is apparent that PA signal strongly depends on the angle of illumination as has been demonstrated by others[1, 2]. Nevertheless, quantification and validation of the reported LED based PAI system is important for accelerating its clinical translation. In addition to the above point, as the reviewers rightly pointed, the novelty of the study is the 3D printed adjustable LED holder which can also be used for orienting laser fiber bundles. The introduction of the manuscript has been updated to describe these sentiments and explicitly state the novelty of the study.

2 & 3) The authors must provide optical absorption and reduced scattering coefficient of the phantom. The percentage weight does not give any indication to the reader about the scattering and absorption of the phantom.

In what way is the phantom mimicking soft tissue. Authors should match both absorption and reduced scattering to well known soft tissue optical properties. How is the lesion mimicking a tumor, what optical properties are mimicked in this lesion? Authors must provide these important information otherwise the claims in this article are questionable.

Thank you very much for pointing out this missing information. The phantoms were prepared to mimic tumor tissue as has been previously reported in literature. We added the relevant quantitative information in the revised version in section 2.2

4) The discussion can be better. Readers will look for a specific angle that is optimal for the focus depth of the transducer. Add discussion on how will the findings vary with tissue having different absorption? Discuss the clinical relevance of this study. Can an adjustable light source be used in any application? Is this study helpful in any way to improve the imaging depth of LED-based PA imaging reported until now?

Thank you for the suggestion. The discussion has been extensively revised. We would like to point that there is no particular optimal angle found in our study especially due to the high divergence beam of the LEDs. The results from the tissue mimicking phantom in scattering media (Figs. 5 and 6) show that PA intensity obtained with such a highly divergent LED illumination weakly depends on the angle of illumination. Overall, our results indicate that deep tissue imaging can be achieved with higher LED illumination angles (closer to being parallel to the imaging plane) and superficial lesion imaging is possible with lower LED illumination angles (closer to being perpendicular to the imaging plane). Given the flexibility, the flexible holder can be used for myriad of clinical applications, from imaging superficial skin lesions to deeply situated lesions such as thyroid or head and neck cancer lesions.

Minor:

1) How is the angle measured in this 3D printed LED array holder? Is this an accurate measurement as there is no reference available in the holder.

A protractor was used for the measurements with the reference being the central axis of the transducer as shown in Fig. 1d. This Information is now added to the manuscript Lines 161-162.

2)  PA intensity and CNR are well-known terms and there is no need to introduce them. Authors can simply use the terms and provide the analysis.

We acknowledge the suggestion. Nevertheless, we feel the equations will provide clarity on our calculations to the readers and hence retained them in the manuscript.

3) Move the concept of measuring divergence to the method section and present only the results in the result section.

We changed the manuscript as per the suggestions. A new section 2.3.2 and a new figure (Fig. 4) is now added to the manuscript.

References

  1. Sivasubramanian, K., et al., Optimizing light delivery through fiber bundle in photoacoustic imaging with clinical ultrasound system: Monte Carlo simulation and experimental validation. J Biomed Opt, 2017. 22(4): p. 41008.
  2. Haisch, C., et al., Combined optoacoustic/ultrasound system for tomographic absorption measurements: possibilities and limitations. Anal Bioanal Chem, 2010. 397(4): p. 1503-10.
  3. Sumit, A., et al. Photoacoustic imaging capabilities of light emitting diodes (LED) and laser sources: a comparison study. in Proc.SPIE. 2020.
  4. Ash, C., et al., Effect of wavelength and beam width on penetration in light-tissue interaction using computational methods. Lasers in medical science, 2017. 32(8): p. 1909-1918.

Reviewer 3 Report

Photoacoustic imaging with affordable light sources such as LEDs has become increasing popular. The manuscript by Kuriakose et al reports an interesting study on the impact of illumination geometry in a LED-based photoacoustic imaging system using a 3D printed flexible light delivery module. With experiments in phantoms with different configurations and optical properties, the authors have demonstrated the weak dependency of the signal strengths for objects at various depths on the illumination angle. These findings provide insightful knowledge for the design and optimisation of future LED-based systems towards preclinical and clinical application. I believe this work would be of great interest to many of us in the photoacoustic research community and beyond. I therefore recommend the publication of this work. However, there are a few minor issues that I would like the authors to consider in the revised version of the manuscript.

  • The flexible light delivery module looks like a useful feature that covers a wide range of angles. However, it would be useful to include angles that are even smaller than 15 degrees, so that there is a larger spatial overlapping between the light illumination and the ultrasound detection. Although a smaller angle tends to generate stronger reflection artefacts as the authors have pointed out.
  • The design and manufacturing details of the flexible light delivery module is missing. It would be important to include such details as this is a central part of the study.
  • What are the fluence at the phantom surfaces for different illumination angles / conditions?
  • The last part of the conclusion section seems a little contradictory to me. The authors first conclude that “there was no significant improvement in CNR for higher illumination 288 angles as the background levels also proportionally increased to the signal from the lesions”, but still recommend the use of different angles for objects are different depths.
  • From previous publications with the same system [Xia W, eta l. Sensors. 2018 May;18(5):1394.], the maximum penetration depths obtained in human volunteers’ studies was between 5 – 10 mm. What is the range of depths that the system is targeting? What angle of illumination would you recommend for this range of depths based on this study.

Author Response

We thank the reviewers for their thoughtful and valuable comments on our manuscript that have significantly improved the manuscript. We heavily revised the manuscript to incorporate the suggestions including adding more figures and supplementary data. A point-by-point response to the reviewers’ comments (in black font) are provided below. The sentences and paragraphs modified as per reviewers’ suggestions are highlighted with a line on the left margin in the manuscript document. We also refer to the new figure numbers in the response below.

  1. The flexible light delivery module looks like a useful feature that covers a wide range of angles. However, it would be useful to include angles that are even smaller than 15 degrees, so that there is a larger spatial overlapping between the light illumination and the ultrasound detection. Although a smaller angle tends to generate stronger reflection artefacts as the authors have pointed out.

Thank you for the comments. We agree with the reviewer that the flexible light delivery is the novel aspect of the manuscript and hence we included the relevant points in introduction and added a separate paragraph in the discussion section. Furthermore, we have used all the angles (0 to 90 degrees) feasible with the light delivery system for our experiments and data analysis. Only three representative angles were shown in Figure 3, however the graphs showcase the data from the entire range of angles. We now included images obtained at all angles in Supplementary data as per reviewer’s suggestion (Figs. S2 and S3).

  1. The design and manufacturing details of the flexible light delivery module is missing. It would be important to include such details as this is a central part of the study.

Thank you. We included a new figure (Fig. 2) and fabrication details of the flexible modular light delivery system in materials and methods section 2.1.2 (Lines 130-145).

  1. What are the fluence at the phantom surfaces for different illumination angles / conditions?

Details added in the revised version Lines 122-129 in the materials and methods section.

  1. The last part of the conclusion section seems a little contradictory to me. The authors first conclude that “there was no significant improvement in CNR for higher illumination angles as the background levels also proportionally increased to the signal from the lesions”, but still recommend the use of different angles for objects are different depths.

Thank you. We modified and clarified the confusion. For the tissue mimicking phantoms that are optically scattering, we observed that the LED illumination angle has minimal effect on CNR of the image. Hence depending on the ergonomics involved and the application, an appropriate angle can be chosen.

  1. From previous publications with the same system [Xia W, eta l. Sensors. 2018 May;18(5):1394.], the maximum penetration depths obtained in human volunteers’ studies was between 5 – 10 mm. What is the range of depths that the system is targeting? What angle of illumination would you recommend for this range of depths based on this study.

We obtained a penetration depth very close to 20 mm (1 cm coupling water + ~2 cm tissue) in tissue mimicking phantom. In Fig. 6d (tissue mimicking phantom), 12, 20 and 28 mm correspond to the distance of the object from transducer. Penetration in this case is 2, 10 and 18 mm respectively. From this data at 10 or 18 mm penetration  depth, there was no significant difference between the various angles examined in this study (Two-way ANOVA statistical test yield high p-value). Hence depending on the application and the contour of the object a convenient angle can be chosen by the operator for deeper lesions.

References

  1. Sivasubramanian, K., et al., Optimizing light delivery through fiber bundle in photoacoustic imaging with clinical ultrasound system: Monte Carlo simulation and experimental validation. J Biomed Opt, 2017. 22(4): p. 41008.
  2. Haisch, C., et al., Combined optoacoustic/ultrasound system for tomographic absorption measurements: possibilities and limitations. Anal Bioanal Chem, 2010. 397(4): p. 1503-10.
  3. Sumit, A., et al. Photoacoustic imaging capabilities of light emitting diodes (LED) and laser sources: a comparison study. in Proc.SPIE. 2020.
  4. Ash, C., et al., Effect of wavelength and beam width on penetration in light-tissue interaction using computational methods. Lasers in medical science, 2017. 32(8): p. 1909-1918.